 Medicine

# Tobacco control policies on cancer prevention in the Eastern Mediterranean Region, 2025–2050: A modeling study

Saeed Nemati[1,2*], Mojtaba Vand Rajabpour[2*], Xiaoshuang Feng[1], Negar Taheri[2], Harriet Rumgay[3], Farrokh Heidari[4], Ebrahim Karimi[4], Sepideh Abdi[2], Mattias Johansson[1], Mahdi Sheikh[1]

**1** Early Detection, Prevention, and Infectious (EPR) Branch, International Agency for Research on Cancer (IARC/WHO), Lyon, France, **2** Cancer Research Centre, Cancer Institute of Iran, Tehran University of Medical Sciences, Tehran, Iran, **3** Cancer Surveillance Branch, International Agency for Research on Cancer (IARC/WHO), Lyon, France, **4** Otorhinolaryngology Research Centre, Tehran University of Medical Sciences, Tehran, Iran

\* nematis@iarc.who.int (SN); rajabpourmojtaba@gmail.com (MVR)

## Abstract

### Background

Despite the implementation of control policies, smoking prevalence remains high in Eastern Mediterranean Region (EMR), and the impact of tobacco control efforts on cancer prevention is unclear. We assessed the potential impact of key policy interventions on tobacco-related cancer incidence in EMR countries from 2025 to 2050.

### Methods and findings

We conducted a modeling study using a country-level historical data to project tobacco smoking prevalence in EMR countries under four scenarios: (i) full implementation of the MPOWER (Monitor, Protection, Offer, Warn, Enforce, and Raise) policy package, (ii) a 10-unit increase in the cigarette affordability index (Higher values of the affordability index indicate that cigarettes are less affordable) (iii) maximized literacy rates (100% adult literacy), and (iv) combined implementation of all three policies. For each scenario, we estimated the Population Attributable Fraction (PAF) of tobacco smoking for 13 cancer types causally linked to tobacco use. The number of preventable cancer cases was calculated using the difference in PAFs between the current and alternative scenarios, referred to as the Potential Impact Fraction (PIF). An estimated 14.3 million tobacco-related cancer cases will occur in the EMR between 2025 and 2050, with over 3 million attributable to current smoking prevalence (PAF = 21.3%; [95% CI: 18.4, 24.6]). Combined implementation of all assessed policies could prevent 442,292 cases (95% CI: 226,987, 660,045) (3.1% of all projected cases; [95% CI: 1.6, 4.6]). The greatest impact was observed in low HDI (Human Development Index) countries, where up to 291,425 (95%

**Data availability statement:** The full analytical code and processed datasets required to reproduce the findings of this study are publicly available on GitHub (https://github.com/nematisaeed17-lab/EMR_tobacco_policy_model/tree/016e882f636f585024c74582132a1f75b4b-b2ca4) and archived in Zenodo (DOI: https://doi.org/10.5281/zenodo.18701105).

**Funding:** The author(s) received no specific funding for this work.

**Competing interests:** The authors have declared that no competing interests exist.

**Abbreviations:** DALYs, disability-adjusted life years; EMR, Eastern Mediterranean Region; GDP, Gross domestic product; GLOBOCAN, Global Cancer Observatory; HDI, Human Development Index; IARC, International Agency for Research in Cancer; PAF, population attributable fraction; PIF, Potential Impact Fraction; RR, relative risk; UNDP, United Nations Development Programme; WHO-FCTC, World Health Organization's Framework Convention on Tobacco Control; CIs, confidence interval.

CI: 198,186, 388,546) cases could be averted. Maximizing literacy showed the highest preventive potential in low ($n = 224,463$; [95% CI: 149,521, 307,386]) and medium HDI ($n = 84,569$; [95% CI: [2,801, 177,317]) countries, while full implementation of MPOWER had the greatest effect in high HDI countries ($n = 11,890$; [95% CI: 8,397, 15,378]). As our main limitation, we assumed a causal relationship between previously implemented policies and concurrent changes, while other potential causes of these changes have not been considered in the current study.

## Conclusion

Strengthening tobacco control policies particularly improving literacy in low HDI countries may potentially contribute to reductions in future cancer burden in EMR.

---

## Author summary

### Why was this study done?

- Tobacco smoking remains highly prevalent in countries of the Eastern Mediterranean Region (EMR) and causes over 100,000 cancer cases each year.

- By 2050, an estimated 14 million cancer cases are expected in EMR countries, and more than 3 million of these could be attributable to tobacco if current trends continue.

- Although most EMR countries have adopted tobacco control policies, the long-term impact of fully implementing these policies on future cancer burden remains unclear.

- This study aimed to estimate how different tobacco control strategies could reduce future tobacco-related cancers in EMR countries.

### What did the researchers do and find?

- We used long-term historical data and modeling techniques to estimate the future impact of three policy strategies: improving literacy rates, increasing tobacco taxation (affordability index), and full implementation of the WHO MPOWER package.

- We estimated that implementing all three strategies together could prevent more than 400,000 new tobacco-related cancer cases by 2050.

- Improving literacy rates showed the largest individual impact, with the potential to prevent over 300,000 cancer cases.

- The greatest potential reductions were observed in lower-income countries, where up to 5% of future tobacco-related cancers could be prevented.

**What do these findings mean?**

- Strengthening tobacco control policies could substantially reduce the future cancer burden in EMR countries.

- Lower-income countries may achieve the largest gains by improving education and literacy, while higher-income countries may benefit more from strengthening taxation and MPOWER implementation.

- These findings are based on modeling assumptions and projections and depend on the accuracy of available data and future policy implementation, which may differ from real-world conditions.

## Introduction

Tobacco smoking remains one of the most major public health challenges worldwide. In 2019, tobacco smoking accounted for an estimated 7.7 million deaths and 20 million disability-adjusted life years (DALYs) globally [1]. In 2012, tobacco smoking led to an economic burden of $1,436 billion worldwide [2]. Tobacco smoking remains the leading cause of cancer globally, that is causally linked to cancers in 13 anatomical sites, including lung, larynx, pharynx, esophageal, oral cavity, bladder, leukemia, stomach, colorectal, cervix, pancreas, liver, and kidney cancer. [3] Given the strong causal link between tobacco smoking and various cancers, even small reductions in the prevalence of tobacco smoking can prevent a large number of future cancers [4].

In Eastern Mediterranean Region (EMR) countries, where smoking tobacco remains highly prevalent [5], particularly among males, the population attributable fraction (PAF) of tobacco smoking for cancer is estimated at 14.6%, resulting in over 104,000 new cancer cases in 2020 [6]. Several factors exacerbate the tobacco epidemic in the EMR, including relatively low cost of cigarettes compared to the rest of the world [7,8], but also the popularity of alternative forms of tobacco products, such as water-pipe [6,9,10]. Indeed, tobacco smoking prevalence is expected to increase in the EMR countries particularly among the youth [5]. This underscores the urgent need for effective tobacco control measures in EMR countries.

In response to the urgent need for tobacco control, 19 out of 22 EMR countries have ratified the World Health Organization's Framework Convention on Tobacco Control (WHO-FCTC) and implemented the MPOWER measures [11]. The MPOWER policy package focus on six key strategies for tobacco control: *(i)* **Monitor**: track tobacco use and prevention policies, *(ii)* **Protect**: protect people from tobacco smoke by enforcing smoke-free laws, *(iii)* **Offer**: offer help to quit tobacco use through various cessation programs, *(iv)* **Warn**: warn about the dangers of tobacco through effective health warnings, *(v)* **Enforce**: enforce bans on tobacco advertising, promotion, and sponsorship, and *(vi)* **Raise**: raise taxes on tobacco to reduce its affordability [11,12]. These strategies collectively aim to reduce tobacco use and its associated health risks at the population level. Over the past decade, considerable improvements in the implementation of the MPOWER package in most EMR countries, with a concurrent 2.8% reduction in the prevalence of tobacco smoking, demonstrating the effectiveness of these initiatives [8]. Despite these efforts, many EMR countries still lag behind in fully implementing anti-tobacco policies.

Using historical data from 2010 to 2020, we previously analyzed various tobacco prevention policies and country-specific prevalence of tobacco smoking across EMR countries. We found that implementation of the MPOWER package, increases in the tobacco affordability index, and rising literacy rates, were associated with changes in smoking prevalence in the region [8]. However, the potential impact of strengthening anti-tobacco policies on cancer incidence in EMR has not been estimated.

We investigated the potential impact of implementing i) tobacco prevention policies, ii) increasing the cost of cigarettes, and iii) improving literacy rates, on the prevention of tobacco-related cancers in EMR over the next 25 years.

## Methods

### Analytical strategy

The current study employed a modeling-based approach using aggregated country-level secondary data. Our study sought to estimate the potential impact of implementing anti-tobacco policies and other measures on the incidence of

tobacco-related cancers in EMR. Three scenarios were considered, highest MPOWER implementation, a 10-unit increase in affordability index, and maximizing literacy level, as well as combined implementation of all policies. As already mentioned MPOWER score has six key components with an overall MPOWER score ranging from 7 to 34. While the maximum score for the 'M' component is 4, the other components each have a maximum score of 5 [13]. Higher values of the affordability index correspond to lower cigarette affordability. The index is measured as the proportion of Gross Domestic Product (GDP) per capita required to purchase 100 packs of cigarettes; therefore, an increase in the index indicates that cigarettes become more expensive relative to income [8]. The rationale for selecting a 10-unit increase in the affordability index was informed by benchmarking against countries with strong tobacco control performance. In most EMR countries, the affordability index remains below 5% of GDP per capita, indicating that cigarettes are relatively affordable. In contrast, in several European countries with well-established tobacco control policies, the affordability index typically ranges between 10% and 15% of GDP per capita [14].

This involved (i) estimating future country-specific prevalence of tobacco smoking based on the observed historical associations between tobacco control measures and changes in tobacco smoking prevalence, (ii) calculating the PAF of tobacco smoking under current smoking prevalence, (iii) calculation the PAFs under each hypothetical scenario of anti-tobacco policies implementation, and (iv) estimating the number of preventable cancer cases under each policy scenario. It should be noted that the statistical approaches used in this study can only identify association, while we assumed a cause-and-effect relationship between previously implemented policies and concurrent changes in tobacco smoking prevalence, but did not consider other potential causes of these changes. We have reported data sources, model inputs, assumptions, and analytical procedures clearly and in accordance with the best practice for modeling studies.

## Estimating future country-specific prevalence of tobacco smoking

Data on tobacco smoking prevalence (Age-standardized prevalence of tobacco smoking, adults aged ≥15 years), MPOWER score, tobacco affordability index, and literacy rate for each country were extracted from the Global Health Observatory Data Repository (EMR) and are presented in S1 Table. Gender-specific tobacco smoking prevalence estimates from 2025 were used as the baseline [15]. The prevalence of tobacco smoking in 2025 was directly extracted from the WHO repository dataset.

To investigate the association between changes in each investigated policy and changes in prevalence of tobacco smoking, we utilized historical data on smoking prevalence, MPOWER score [16], affordability index [14,17], and literacy rate in the past decade (2010, 2012, 2014, 2016, 2018, and 2020) for each EMR country [18]. We excluded five countries including Sudan, Somalia, Libya, Syria, and Djibouti due to lack of recent data on prevalence of tobacco smoking. Smoking status variable and literacy rate were age-standardized prior to analysis, consistent with international reporting. However, no additional standardization was applied for other variables such as MPOWER score, and affordability index before regression. Policy related exposures such as MPOWER and affordability index were retained in their original measurement scales, as these variables represent policy implementation level rather than continuous biological measures. Fixed-term linear regression models were used to investigate the association between changes in the assessed policies and changes in the prevalence of tobacco smoking using the following equation. The regression models were not population weighted. Given the ecological study design, each country was treated as a single unit of observation with equal contribution to the analysis (Equation 1).

$$\text{Equation 1} : \text{Prevalence}_{it} = \beta_0 + \beta_1 \cdot \text{MPOWER}_{i,t} + \text{Country}_i + \text{Year}_t + \epsilon_{it}$$

Here, $Prevalence_{i,t}$ denotes the age-standardized prevalence of tobacco smoking in country $i$ at time $t$, and $MPOWER_{i,t}$ indicates the MPOWER score for the same country and year. Country and year were included as fixed-effects. The country fixed effect controls for unobserved time-invariant heterogeneity across countries, while the year fixed effect captures

temporal shocks or factors common to all countries. Smoking prevalence was modeled as an absolute percentage (0–100), and regression coefficients therefore represent absolute percentage-point changes associated with a one-unit increase in the independent variables.

The effects of the affordability index and literacy rate on smoking prevalence were modeled using similar fixed-effects specifications: (Equations 2, and 3)

$$\text{Equation 2}: \text{Prevalence}_{it} = \beta_0 + \beta_1 \text{ affordability index}_{it} + \text{Country}_i + \text{Year}_t + \epsilon_{it}$$

$$\text{Equation 3}: \text{Prevalence}_{it} = \beta_0 + \beta_1 \text{ literacy rate}_{it} + \text{Country}_i + \text{Year}_t + \epsilon_{it}$$

Finally, a multiple fixed-effects regression model was employed to evaluate the combined effects of MPOWER scores, affordability index, and literacy rates on tobacco smoking prevalence (Equation 4). These analyses were stratified by gender, and the regression results are reported in S2 Table. For transparency, we have also reported model fit statistics for all fixed-effects models, including within, between, and overall $R^2$ values (S2 Table). Multiple collinearities among covariates were assessed using variance inflation factor and no evidence of multicollinearity was observed for both men and women. Conceptual framework linking different policy measures to prevalence of tobacco smoking and cancer incidence is presented in Fig 1.

$$\text{Equation 4}: \text{Prevalence}_{it} = \beta_0 + \beta_1 \cdot \text{MPOWER}_{i,t} + \beta_2 \text{ affordability index}_{it} + \beta_3 \text{ literacy rate}_{it} + \text{Year}_t + \epsilon_{it}$$

To estimate the reduction in tobacco smoking prevalence in each EMR country under different scenarios, we first calculated the difference between the current values of the MPOWER score, literacy rate, and affordability index and their respective targets. These targets were defined as the maximum achievable MPOWER score, the maximum literacy rate, and a 10-unit increase in the affordability index. Using coefficients from the fixed-term regression model, we then

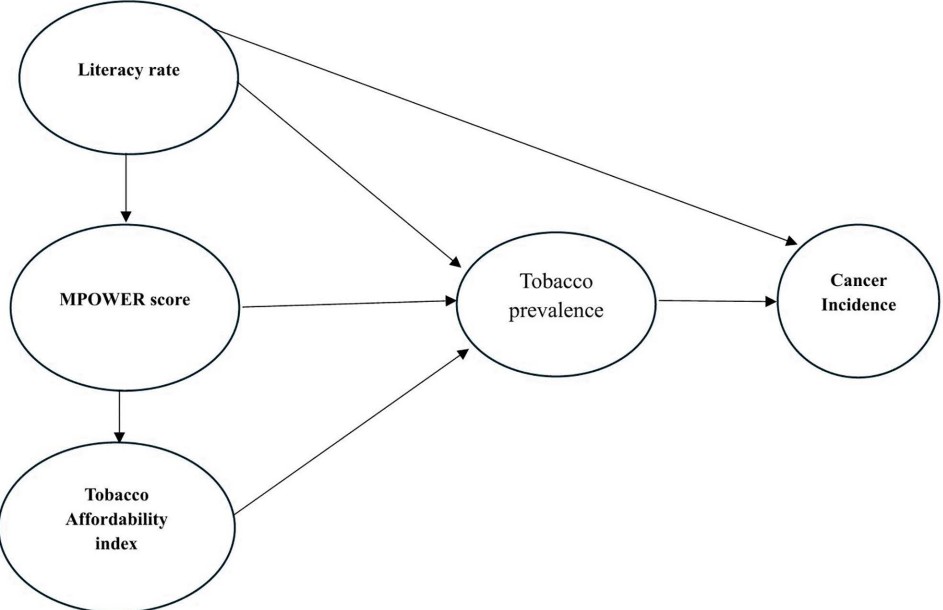

**Fig 1. Conceptual framework linking literacy, tobacco control policies, smoking prevalence, and cancer incidence.**

estimated the reduction in smoking prevalence for each policy by multiplying the policy gap (the difference between the current value and the target) by the corresponding regression coefficient. For the scenario assessing the combined implementation of all policies, the reductions from each policy (regression coefficients from multiple-adjusted model were used) were summed to calculate the total reduction in smoking prevalence. Additionally, we performed a gender-specific analysis, calculating the reductions separately for men and women in each EMR country. Gender-specific models were conducted by estimating separate regression model for men and women. Difference between men and women were therefore describe qualitatively, and formal statistical comparisons were not the primary objective of the study. To avoid implausible negative prevalence values in low-prevalence settings, projected smoking prevalence was constrained to a minimum value of zero in all alternative scenarios.

We then estimated the gender-specific prevalence of tobacco smoking under each alternative scenario for every EMR country by subtracting the projected reduction associated with each policy from the current prevalence. For example, the prevalence under the scenario of full MPOWER implementation was calculated as follows: (Equation 5)

**Equation 5** : Prevalence Under full MPOWER = $\text{Prevalence}_{x2025} - \text{Reduction from MPOWER}$

Gender-specific Current prevalence of tobacco smoking, prevalence of tobacco smoking with the highest MPOWER, prevalence with a 10-unit increases in affordability index, prevalence with maximizing literacy rate, and prevalence of tobacco smoking under all policies combined scenario in each EMR country are presented in S3 Table.

## Estimating the number of preventable cancer cases under each policy scenario

To estimate the number of tobacco-related cancers for each year, we used the projected crude, gender-specific incidence of cancers for the years 2025–2050, as provided by the Global Cancer Observatory (GLOBOCAN) of the International Agency for Research in Cancer (IARC) [19]. Our analysis specifically included cancer sites that have been identified by IARC monographs to be causally linked to tobacco smoking [3,20], including lung, larynx, pharynx, esophageal, oral cavity, bladder, leukemia, stomach, colorectal, cervix, pancreas, liver, and kidney cancer. Cancer incidence projections were obtained from GLOBOCAN, which provides estimates of the absolute number of incident cancer cases at 5-year intervals (2025, 2030, 2035, 2040, 2045, and 2050). To derive annual estimates, we calculated the difference in projected case numbers between two consecutive 5-year time points and then divided this difference by five, assuming a constant linear increase within each 5-year interval. The resulting annual increments were subsequently applied to estimate cancer incidence for intermediate years. The total number incident cancers in each country was estimated as the sum of all the incident cancers in each year from 2025 to 2050. The same procedure was applied to each tobacco-related cancer separately to calculate the number of all incident cancers associated with tobacco smoking (S4 Table).

**Relative risk of cancers in relation to tobacco smoking.** The relative risk (RR) utilized in the current study was extracted from the latest published meta-analysis on the association between current cigarette smoking and each included cancer type (S5 Table).

**Preventable number of cancers.** The calculation of the PAF involved gender-specific PAFs under the current prevalence and all other alternative scenarios, which were computed using Levin's equation (Equation 6).

**Equation 6** : $\text{PAF} = \frac{P(RR-1)}{1 + P(RR-1)}$

Subsequently, we multiplied the estimated PAF in each scenario by the number of all incident cancers to calculate the number of cancers attributable to tobacco smoking. Finally, we subtracted the PAF in the current prevalence scenario from the PAF of all attributable cancers under each scenario separately and multiplied it into the number of predicted cancers to calculate the number of avoidable cancers under each scenario (Equation 7).

**Equation 7** : N of preventable cancers = $(\text{PAF}_{\text{with current prevalence}} - \text{PAF}_{\text{with the alternative scenario}}) * \text{N of predicted cancers}$

We referred to the difference between the PAF of the current prevalence and the PAF of alternative scenarios as the Potential Impact Fraction (PIF) which was proportion of cancer that could be prevented attributed to each anti-tobacco policy. Uncertainty for PAF/PIF was assessed using a nonparametric bootstrap procedure with 1,000 replications at the country-level. The 95% confidence interval (Cis) was calculated from the empirical distributions of the bootstrap estimates.

We categorized the countries into three groups based on their Human Development Index (HDI) to conduct a subgroup analysis aimed at identifying the most effective anti-tobacco policies tailored to their socioeconomic status. The **low HDI group** included Afghanistan, Pakistan, Iraq, Morocco, and Yemen. The **medium HDI group** comprised Egypt, Iran, Lebanon, Tunisia, and Jordan, while the **high HDI group** consisted of Bahrain, Kuwait, Oman, Qatar, Saudi Arabia, and the UAE. HDI data were sourced from the United Nations Development Programme (UNDP) [21]. HDI categories were used only for post-estimation aggregation of projected cancer cases and were not included in the regression models.

### Sensitivity analysis

Sensitivity analyses incorporating two alternative cancer projection scenarios were added. In these analyses, we assumed that future cancer incidence projections were 10% higher and 10% lower than the GLOBOCAN estimates, to assess the robustness of our findings to uncertainty in cancer burden projections. We also evaluated the presence of interactions between policy measures.

## Results

### Tobacco smoking prevalence, MPOWER scores, affordability index, and literacy rates in EMR countries

According to S1 Table, tobacco smoking prevalence was higher among men than in women in each evaluated country. The highest observed prevalence in 2025 for men was observed in Jordan (58.4%), Egypt (51.2%), and Lebanon (43.3%). For women, Lebanon (25.7%), Jordan (14.1%), and Yemen (5.9%) were the leading countries regarding prevalence of tobacco smoking. MPOWER scores ranged from 20 in Oman and Afghanistan to 31 in Iran. The affordability index for most EMR countries was less than 5% of GDP per capita (S1 Table).

### Projected incident cancers attributable to the current prevalence of tobacco smoking by country

According to current pattern of cancer incidence, we estimated that 14,308,033 tobacco-related cancers will occur in EMR countries over a 25-year period from 2025 to 2050. Iran (*n*: 3,549,157), Egypt (*n*: 3,136,020), and Pakistan (*n*: 3,092,670) have the highest projected number of tobacco-related cancers in the region. If the current prevalence of tobacco smoking continues, we estimated that 3,050,928 (95% CI: 2,638,201, 3,517,213) incident cancer cases can be attributed to tobacco over the next 25 years (PAF overall: 21.3%, [95% CI: 18.4, 24.6]). The highest PAF for cancers related to tobacco smoking was observed in Lebanon (PAF = 38.3, [95% CI: 34.6, 42.0]), Jordan (PAF = 36.5, [95% CI: 33.3, 39.7]), and Tunisia (PAF = 34.3, [95% CI: 30.9, 37.8]). The highest number of projected cancers attributable to tobacco smoking was in Egypt (*n* = 838,909, [95% CI: 758,695, 942,620]), Pakistan (*n* = 648,419, [95% CI:538,748, 760,048]), and Iran (*n* = 459,695, [95% CI: 370,854, 566,185]) (S4 Table).

### Projected preventable cancers with the implementation of each tobacco control policy by country

Implementation of the MPOWER measures at the highest level was projected to prevent 154,209 (95% CI: 92,196, 208,033) incident cancer cases (PIF: 1.1, [95% CI: 0.6, 1.5]) by 2050 in EMR countries, with the highest impact observed for Pakistan with 48,907 (95% CI: 35,420, 62,394), and Afghanistan with 13,674 (95% CI: 12,029, 15,319) projected preventable cancer cases. The highest proportions of preventable cancers were found in Afghanistan (PIF = 2.9, [95% CI: 2.6, 3.3]), Oman (PIF = 2.2, [95% CI: 1.9, 2.5]), and Morocco (PIF = 1.8, [95% CI: 1.3, 2.3]) (Table 1 and Fig 2).

Increasing the tobacco affordability index, as a second alternative scenario, was projected to prevent 0.8% (95% CI: 0.6, 0.9) of all incident tobacco-related cancers in the EMR (n = 108,703; [95% CI: 79,381, 127,622]). More than half of these preventable cancers were estimated to occur in Pakistan and Iran (Table 1 and Fig 2).

Maximizing the population literacy rate, as the third anti-tobacco scenario, was projected to prevent the highest number of cancer cases among the assessed anti-tobacco policies, with a potential to prevent over 300,000 cancer cases by 2050 in the EMR countries (N = 311,107, [95% CI: 152,420, 499,504]). This was mostly driven by Pakistan with more than 175,000 preventable cancers (N = 143,125, [95% CI: 100,108, 186,143]), followed by Afghanistan, Egypt, and Iran. We estimate that 2.2% (95% CI: 1.1, 3.5) of tobacco-related cancers could be potentially prevented by maximizing literacy rate in this region. The highest proportions were observed in Afghanistan (PIF = 9.5, [95% CI: 8.4, 10.6]), Pakistan (PIF = 4.6, [95% CI: 3.2, 6.0]), Yemen (PIF = 2.4, [95% CI = 1.5, 3.3]), and Morocco (PIF = 2.1, [95% CI: 0.4, 4.0]) (Table 1 and Fig 2).

Combined implementation of all assessed anti-tobacco policies was projected to prevent 442,292 (95% CI: 226,987, 660,045) of all incident tobacco-related cancers, equivalent to a total 3.1% (95%: 1.6, 4.6) of new cancer cases in EMR countries, over the next 25 years. The highest PIF for the combined implementation of all policies was 10.9% (95% CI: 9.7, 12.2) which was observed for Afghanistan, followed by Pakistan (PIF = 5.7, [95% CI = 4.0, 7.3]), Yemen (PIF = 3.7, [95% CI: 2.6, 4.9]), Morocco (PIF = 3.5, [95% CI: 1.3, 5.7]), and Tunisia (PIF = 2.4, [95% CI: 0.8, 4.6]) (Table 1 and Fig 2).

### Projected incident cancers attributable to the current prevalence of tobacco smoking by cancer site

Lung cancer (PAF = 53.6%, [95% CI: 48.6, 58.9]), larynx cancer (PAF = 52.3%, [95% CI: 47.0, 57.9]), pharynx cancer (PAF = 43.3%, [95% CI: 38.3, 48.4]), and bladder cancer (PAF = 34.6%, [95% CI: 30.1, 39.7]) are projected to have the highest proportions of cancers related to current tobacco smoking in the EMR (S6 Table).

### Projected preventable cancers with the implementation of each tobacco control policy by Cancer site

The combined implementation of all assessed anti-tobacco policies was projected to prevent 8.0% (95% CI: 4.6, 11.4) of projected pharynx cancers, 6.8% (95% CI: 3.2, 11.1) of larynx cases, and 6.8% (95% CI: 5.0, 8.6) of new incident oral cancer cases over the investigated period in EMR countries. Lung cancer (PIF = 6.0, [95% CI: 2.6, 10.3]), bladder cancers (PIF = 4.0, 95% [CI: 1.7, 6.1]), and esophageal cancer (PIF = 3.7, [95% CI: 2.5, 4.9]) were other cancer sites with notable contributions (Fig 3) (Table 2).

### Number and proportion of preventable cancer cases by socioeconomic status

We conducted a subgroup analysis based on the HDI of EMR countries, examining the numbers and proportions of preventable cancers under the implementation of different anti-tobacco policies for countries with high, medium, and low HDI. The proportions of preventable projected cancers under the combined implementation of all assessed anti-tobacco policies were highest in countries with low HDI (PIF = 5.1%, [95% CI: 3.5, 6.8]), and lowest in countries with high HDI (PIF = 1.4%, [95% CI: 0.4, 2.7]). Our analysis showed that the combined implementation of all policies is projected to prevent 291,425 (95% CI: 198,186, 388,546) cancer cases in countries with low HDI, 135,638 (95% CI: 23,919, 241,378) cases in countries with medium HDI, and 15,228 (95% CI: 4,883, 30,121) cases in countries with high HDI (S7 Table).

Assessing the impact of each individual anti-tobacco policies by HDI, revealed differences by HDI levels; maximizing the MPOWER score (PIF = 1.1, [95% CI: 0.8, 1.4]) was identified as the most effective policy for countries with high HDI, while maximizing literacy rates was the greatest preventive impact for countries with low HDI (PIF = 3.9, [95% CI: 2.6, 5.4]) (S7 Table).

**Table 1. Number of projected preventable cancers by 2050, that could be achieved though highest MPOWER implementation, a 10-unit increase in tobacco affordability index, maximizing literacy rate, and combined implementations of all policies in EMR countries.**

| Both genders | Preventable cancer by highest MPOWER | | Preventable cancer by a 10-unit increases in tobacco affordability index | |
| --- | --- | --- | --- | --- |
| Country | PIF (95% CI) | N of cancer (95% CI) | PIF (95% CI) | N of cancer (95% CI) |
| Afghanistan | 2.9 (2.6, 3.3) | 13,674 (12,029, 15,319) | 1.0 (0.8, 1.1) | 4,540 (3,947, 5,133) |
| Bahrain | 1.3 (0.9,1.7) | 441 (308, 573) | 0.8 (0.6, 0.9) | 257 (211, 303) |
| Egypt | 0.6 (0.0, 0.9) | 18,696 (1,016, 28,938) | 0.4 (0.1, 0.5) | 13,936 (1,616, 17,187) |
| Iran | 0.6 (0.2, 0.9) | 20,708 (8,470, 32,831) | 0.9 (0.8, 1.1) | 33,075 (28,578, 37,294) |
| Iraq | 1.1 (0.7, 1.5) | 7,525 (4,651, 10,284) | 0.6 (0.5, 0.8) | 4,411 (3,380, 5,373) |
| Jordan | 0.3 (0.0, 0.7) | 793 (16, 1,848) | 0.4 (0.0, 0.5) | 942 (98, 1,307) |
| Kuwait | 1.7 (1.3, 2.0) | 2,355 (1,866, 2,844) | 0.6 (0.5, 0.8) | 905 (739, 1,071) |
| Lebanon | 0.8 (0.3, 1.3) | 1,253 (530, 1,975) | 0.4 (0.3, 0.6) | 653 (391, 915) |
| Morocco | 1.8 (1.3, 2.3) | 19,329 (13,736, 24,388) | 0.9 (0.7, 1.1) | 9,734 (7,587, 11,515) |
| Oman | 2.2 (1.9, 2.5) | 1,695 (1,460, 1,927) | 0.8 (0.7, 0.9) | 592 (514, 668) |
| Pakistan | 1.6 (1.1, 2.0) | 48,907 (35,420, 62,394) | 0.9 (0.8, 1.1) | 28,603 (23,512, 33,694) |
| Qatar | 1.2 (0.9, 1.5) | 379 (274, 484) | 0.8 (0.7, 0.9) | 254 (218, 290) |
| Saudi Arabia | 0.7 (0.4, 1.0) | 4,781 (2,651, 6,910) | 0.7 (0.5, 0.8) | 4,529 (3,794, 5,265) |
| Tunisia | 1.3 (0.8, 1.8) | 5,365 (3,126, 7,343) | 0.7 (0.5, 0.8) | 2,779 (1,952, 3,468) |
| United Arab Emirates | 1.8 (1.5, 2.1) | 2,239 (1,839, 2,639) | 0.9 (0.8, 1.0) | 1,151 (1,013, 1,289) |
| Yemen | 1.7 (1.3, 2.0) | 6,068 (4,803, 7,333) | 0.6 (0.5, 0.8) | 2,340 (1,830, 2,850) |
| EMRO | 1.1 (0.6, 1.5) | 154,209 (92,196, 208,033) | 0.8 (0.6, 0.9) | 108,703 (79,381, 127,622) |
| | Preventable cancer by maximizing literacy rate | | Preventable cancers by combined implementation of all policies | |
| Country | PIF (95% CI) | N of cancer (95% CI) | PIF (95% CI) | N of cancer (95% CI) |
| Afghanistan | 9.5 (8.4, 10.6) | 44,656 (39,598, 49,713) | 10.9 (9.7, 12.2) | 51,398 (45,398, 57,398) |
| Bahrain | 0.2 (0.0, 1.7) | 51 (1, 577) | 1.6 (0.4, 3.3) | 522 (136, 1,105) |
| Egypt | 1.4 (0.0, 2.6) | 44,254 (512, 81,066) | 1.8 (0.0, 3.2) | 55,746 (917, 98,820) |
| Iran | 1.0 (0.0, 2.1) | 34,564 (1,654, 76,081) | 1.9 (0.5, 3.2) | 66,729 (19,218, 112,774) |
| Iraq | 0.7 (0.1, 2.3) | 4,941 (378, 15,605) | 1.9 (0.8, 3.6) | 12,857 (5,179, 24,919) |
| Jordan | 0.1 (0.0, 1.7) | 182 (18, 4,375) | 0.5 (0.0, 2.3) | 1,360 (76, 6,073) |
| Kuwait | 0.3 (0.0, 1.6) | 375 (11, 2,206) | 1.9 (0.8, 3.4) | 2,639 (1,098, 4,725) |
| Lebanon | 0.5 (0.0, 2.0) | 782 (35, 2,986) | 1.3 (0.3, 3.0) | 2,001 (423, 4,642) |
| Morocco | 2.1 (0.4, 4.0) | 22,948 (4,077, 43,698) | 3.5 (1.3, 5.7) | 38,278 (13,941, 62,113) |
| Oman | 0.2 (0.0, 1.3) | 158 (8, 1,020) | 2.0 (0.7, 3.3) | 1,552 (509, 2,534) |
| Pakistan | 4.6 (3.2, 6.0) | 143,125 (100,108, 186,143) | 5.7 (4.0, 7.3) | 175,302 (124,391, 226,213) |
| Qatar | 0.9 (0.1, 2.1) | 290 (26, 680) | 2.2 (0.7, 3.6) | 684 (233, 1,134) |
| Saudi Arabia | 0.1 (0.0, 1.3) | 986 (42, 8,726) | 1.0 (0.3, 2.4) | 7,286 (1,945, 16,498) |
| Tunisia | 1.2 (0.1, 3.1) | 4,787 (582, 12,808) | 2.4 (0.8, 4.6) | 9,803 (3,285, 19,068) |
| United Arab Emirates | 0.2 (0.0, 1.3) | 216 (9, 1,592) | 2.1 (0.8, 3.3) | 2,545 (962, 4,126) |
| Yemen | 2.4 (1.5, 3.4) | 8,794 (5,360, 12,228) | 3.7 (2.6, 4.9) | 13,591 (9,277, 17,905) |
| EMRO | 2.2 (1.1, 3.5) | 311,107 (152,420, 499,504) | 3.1 (1.6, 4.6) | 442,292 (226,987, 660,045) |

PIF = Potential Impact Fraction; EMR = Eastern Mediterranean Region; CI = Confidence Interval.

For countries with high HDI, increasing tobacco affordability index and maximizing literacy rate were projected to prevent 0.7% (95% CI: 0.6, 0.8), and 0.2% (95% CI: 0.0, 1.3) of all tobacco-related cancers, respectively. Across all policies, the largest potential improvement was observed in low HDI countries (Fig 4). More details are provided in S7 Table.

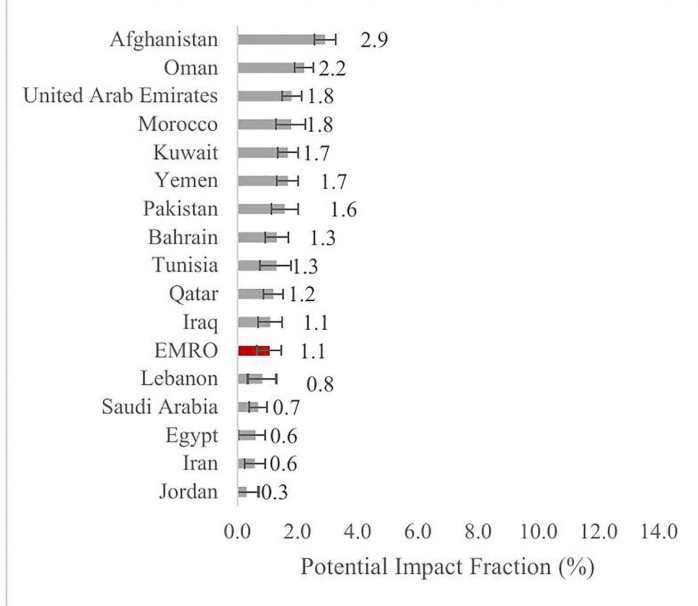

**A: MPOWER**

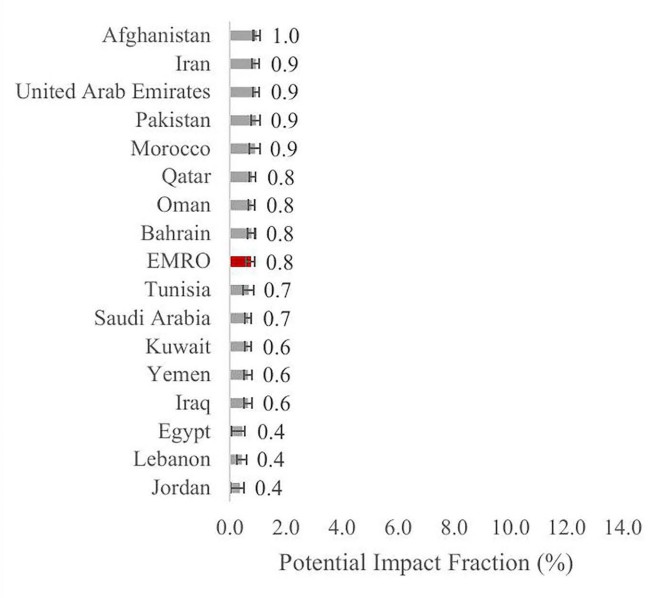

**B: Affordability index**

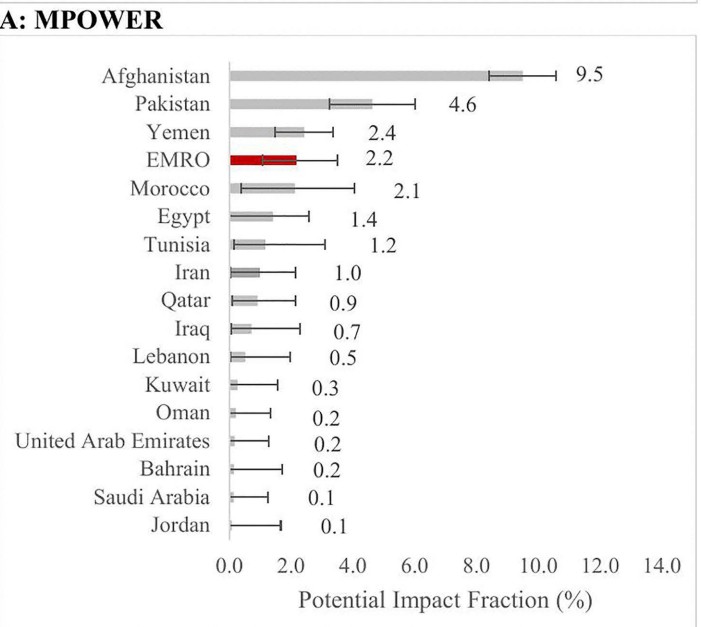

**C: Maximizing literacy level to 100%**

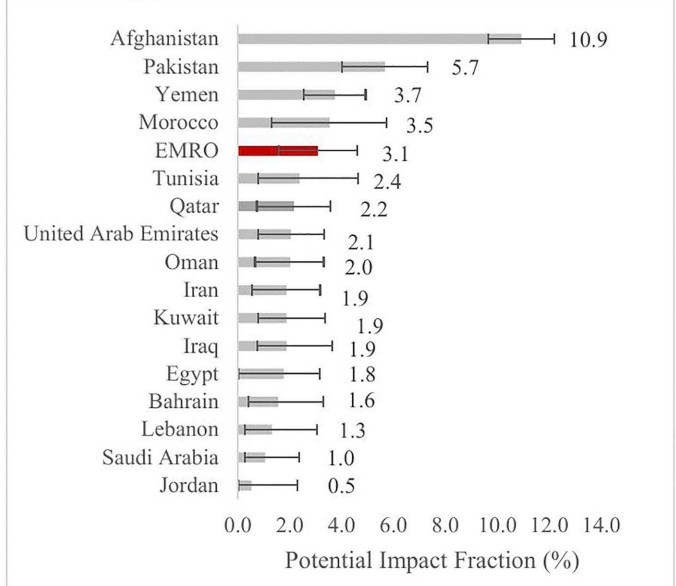

**D: All policies combined**

**Fig 2. Population proportion of preventable cancer in the EMR countries through implementation of each anti-tobacco policies until 2050 stratified by country. A:** Proportion of potentially preventable cancers under highest implementation of MPOWER, **B:** proportion of potentially preventable cancers under a 10-unit escalation of affordability index, **C:** proportion of potentially preventable cancers under maximizing literacy rate up to 100%, **D:** proportion of potentially preventable cancers under all combined policies, The error bars represent the upper and lower bounds of the 95% confidence intervals for the proportion of preventable cancers under each scenario. EMR: Eastern Mediterranean Region.

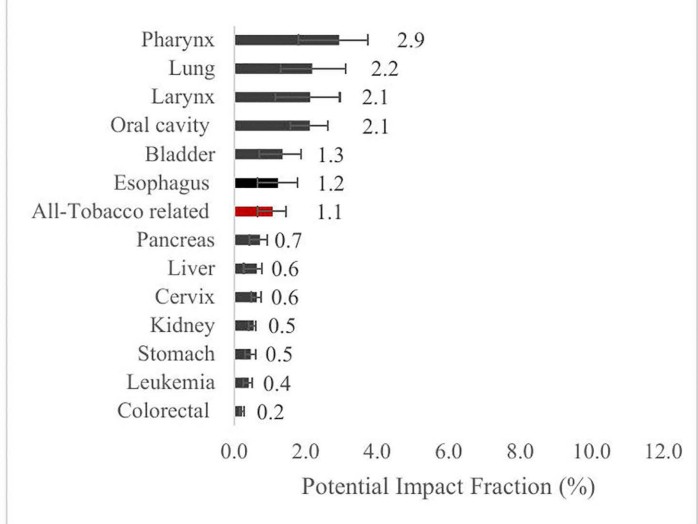

**A: MPOWER**

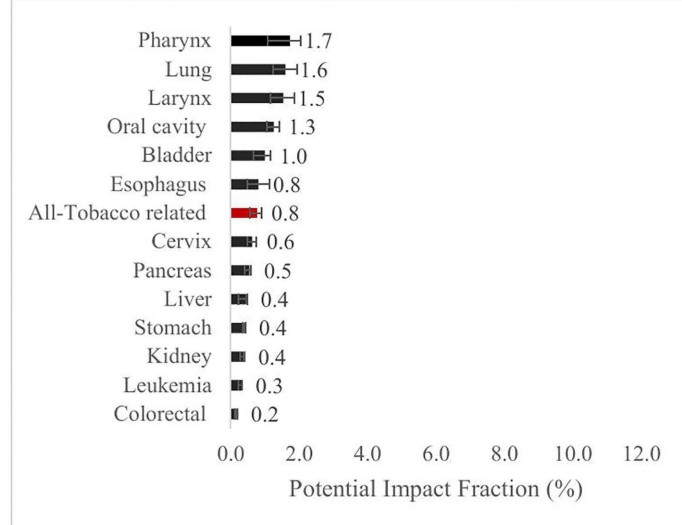

**B: Affordability index**

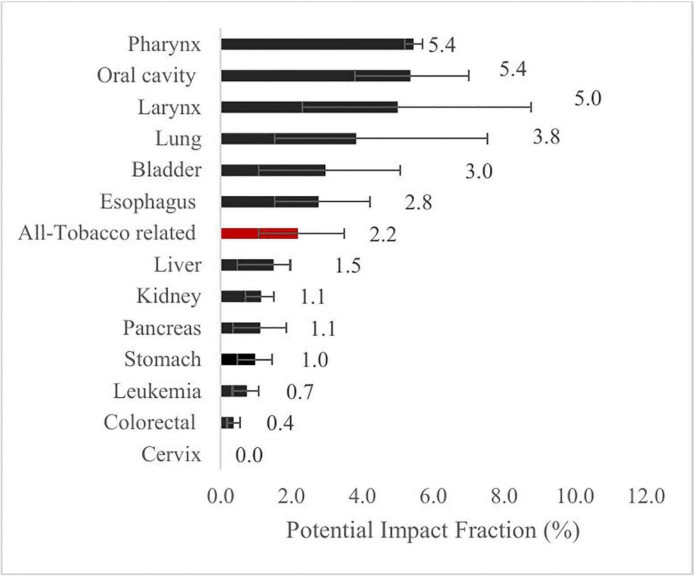

**C: Maximizing literacy level to 100%**

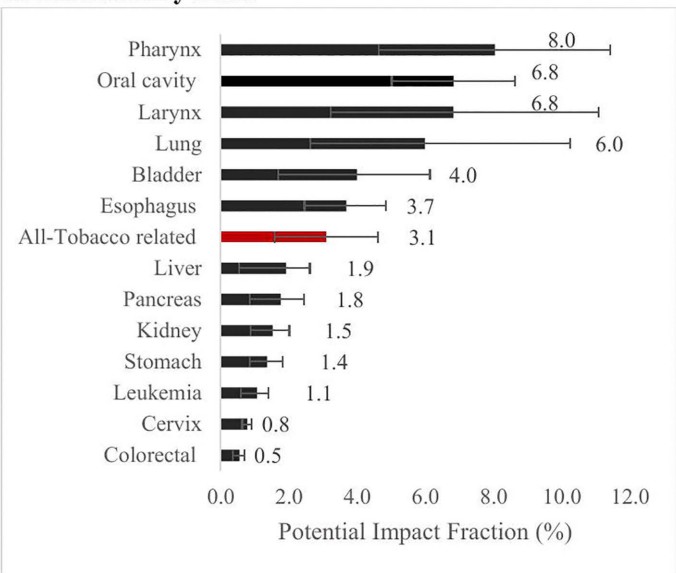

**D: All policies combined**

**Fig 3. Population proportion of avoidable cancer in the EMR countries through implementation of each anti-tobacco policies until 2050 stratified by cancer site. A:** Proportion of potentially preventable cancers under highest implementation of MPOWER, **B:** proportion of potentially preventable cancers under a 10-unit escalation of affordability index, **C:** proportion of potentially preventable cancers under maximizing literacy rate up to 100%, **D:** proportion of potentially preventable cancers under all combined policies, The error bars represent the upper and lower bounds of the 95% confidence intervals for the proportion of preventable cancers under each scenario. EMR: Eastern Mediterranean Region.

## Sensitivity analysis

We performed a sensitivity analysis assuming cancer incidence to be either 10% higher or 10% lower than the number of tobacco-related cancers projected by GLOBOCAN, to account for potential underestimation or overestimation in our baseline projections. Under the higher-incidence scenario, the number of preventable cancers under the combined policy scenario was estimated at 487,000 (95% CI: 250,000–726,000). In contrast, under the lower-incidence scenario, the

**Table 2. Number of avoidable cancers due to highest MPOWER, a 10-unit increase in affordability index, maximizing level of literacy, and all combined policies in EMR countries (2025-2050) stratified by cancer site.**

| Both genders | Preventable cancer by highest MPOWER | | Preventable cancer by a 10-unit increases in tobacco affordability index | |
|---|---|---|---|---|
| | PIF (95% CI) | N of cancer (95% CI) | PIF (95% CI) | N of cancer (95% CI) |
| Lung | 2.2 (1.3, 3.1) | 47,109 (27,963, 67,278) | 1.6 (1.2, 1.9) | 34,482 (26,817, 41,987) |
| Larynx | 2.1 (1.1, 3.0) | 11,050 (5,974, 15,370) | 1.5 (1.2, 1.9) | 7,960 (6,049, 9,655) |
| Esophagus | 1.2 (0.6, 1.8) | 9,248 (4,951, 13,519) | 0.8 (0.5, 1.1) | 6,162 (3,619, 8,598) |
| Pharynx | 2.9 (1.8, 3.7) | 9,262 (5,646, 11,796) | 1.7 (1.1, 2.1) | 5,443 (3,425, 6,470) |
| Oral cavity | 2.1 (1.6, 2.6) | 18,933 (14,075, 23,598) | 1.3 (1.0, 1.4) | 11,267 (9,412, 12,781) |
| Stomach | 0.5 (0.3, 0.6) | 7,830 (5,114, 10,243) | 0.4 (0.3, 0.4) | 6,696 (5,892, 7,145) |
| Colorectal | 0.2 (0.2, 0.3) | 4,539 (3,175, 5,618) | 0.2 (0.1, 0.2) | 3,297 (2,405, 3,933) |
| Liver | 0.6 (0.3, 0.8) | 12,801 (5,392, 15,572) | 0.4 (0.2, 0.5) | 9,113 (4,472, 9,656) |
| Pancreas | 0.7 (0.4, 0.9) | 4,244 (2,538, 5,482) | 0.5 (0.4, 0.6) | 3,229 (2,419, 3,504) |
| Leukemia | 0.4 (0.2, 0.5) | 3,197 (1,984, 4,061) | 0.3 (0.2, 0.3) | 2,370 (1,851, 2,518) |
| Bladder | 1.3 (0.7, 1.9) | 20,301 (10,626, 28,257) | 1.0 (0.7, 1.2) | 14,997 (10,012, 17,342) |
| Kidney | 0.5 (0.4, 0.6) | 2,411 (1,828, 2,707) | 0.4 (0.3, 0.4) | 1,654 (1,270, 1,815) |
| Cervix | 0.6 (0.5, 0.8) | 3,283 (2,556, 3,963) | 0.6 (0.5, 0.8) | 3,283 (2,556, 3,963) |
| All Tobacco-related | 1.1 (0.6, 1.5) | 154,209 (92,196, 208,033) | 0.8 (0.6, 0.9) | 108,703 (79,381, 127,622) |
| | Preventable cancer by maximizing literacy rate | | Preventable cancers by combined implementation of all policies | |
| | PIF (95% CI) | N of cancer (95% CI) | PIF (95% CI) | N of cancer (95% CI) |
| Lung | 3.8 (1.5, 7.5) | 82,697 (33,223, 162,805) | 6.0 (2.6, 10.3) | 129,499 (57,042, 221,900) |
| Larynx | 5.0 (2.3, 8.8) | 25,975 (11,973, 45,543) | 6.8 (3.2, 11.1) | 35,477 (16,786, 57,699) |
| Esophagus | 2.8 (1.5, 4.2) | 21,017 (11,575, 32,166) | 3.7 (2.5, 4.9) | 28,074 (18,737, 37,043) |
| Pharynx | 5.4 (5.2, 5.7) | 17,157 (16,344, 17,993) | 8.0 (4.6, 11.4) | 25,332 (14,649, 36,007) |
| Oral cavity | 5.4 (3.8, 7.0) | 48,052 (34,077, 62,859) | 6.8 (5.0, 8.6) | 61,309 (45,021, 77,552) |
| Stomach | 1.0 (0.5, 1.5) | 16,414 (7,994, 24,650) | 1.4 (0.8, 1.8) | 22,935 (14,277, 30,669) |
| Colorectal | 0.4 (0.2, 0.5) | 7,387 (3,759, 10,997) | 0.5 (0.3, 0.7) | 11,172 (7,089, 14,389) |
| Liver | 1.5 (0.5, 2.0) | 30,339 (9,762, 40,011) | 1.9 (0.5, 2.6) | 38,827 (11,150, 53,153) |
| Pancreas | 1.1 (0.3, 1.9) | 6,595 (1,988, 10,976) | 1.8 (0.9, 2.4) | 10,368 (5,074, 14,380) |
| Leukemia | 0.7 (0.3, 1.1) | 5,907 (2,646, 8,627) | 1.1 (0.6, 1.4) | 8,460 (4,629, 11,184) |
| Bladder | 3.0 (1.1, 5.1) | 44,519 (16,022, 76,258) | 4.0 (1.7, 6.1) | 60,027 (25,327, 92,377) |
| Kidney | 1.1 (0.7, 1.5) | 5,048 (3,057, 6,619) | 1.5 (0.9, 2.0) | 6,752 (3,905, 8,929) |
| Cervix | 0.0 (0.0, 0.0) | 0 (0, 0) | 0.8 (0.6, 0.9) | 4,062 (3,301, 4,763) |
| All Tobacco-related | 2.2 (1.1, 3.5) | 311,107 (152,420, 499,504) | 3.1 (1.6, 4.6) | 442,292 (226,987, 660,045) |

PIF = Potential Impact Fraction; EMR = Eastern Mediterranean Region; CI = Confidence Interval.

estimated number decreased to 398,000 (95% CI: 204,000–594,000) under the same tobacco prevention scenario. The results are presented by country and cancer site (S8–S11 Tables).

We also performed another sensitivity analysis evaluating interaction between different policy measures. Our analysis identified a statistically significant interaction between literacy rate and MPOWER score, indicating that these factors jointly modify smoking behavior. Specifically, the effect of the MPOWER score was attenuated at higher literacy levels, and this effect modification was observed in both men and women. The corresponding regression results and estimates of preventable cancer cases are provided in S12–S14 Tables.

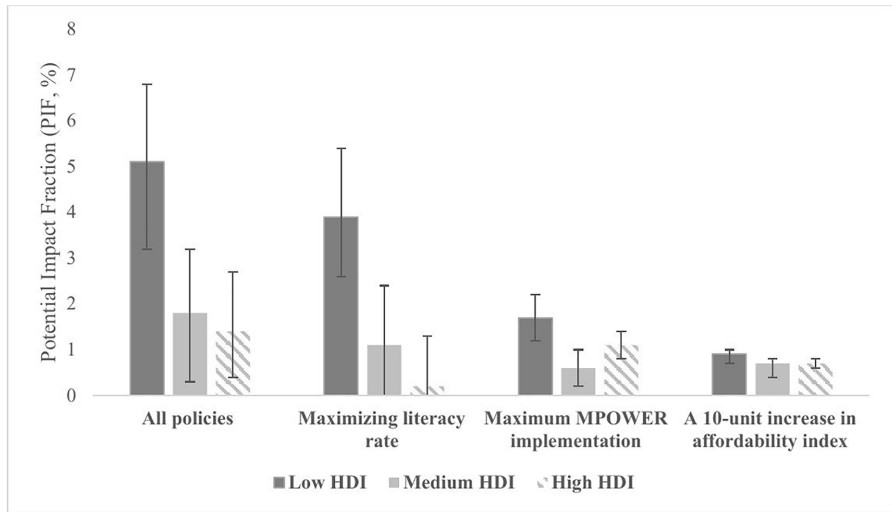

**Fig 4. Estimated proportion of preventable tobacco-related cancers under each prevention scenario, stratified by Human Development Index of EMR countries, 2025–2050.** HDI: Human Development Index, EMR: Eastern Mediterranean Region, PIF = Potential Impact Fraction.The **low HDI group** included Afghanistan, Pakistan, Iraq, Morocco, and Yemen. The **medium HDI group** comprised Egypt, Iran, Lebanon, Tunisia, and Jordan, while the **high HDI group** consisted of Bahrain, Kuwait, Oman, Qatar, Saudi Arabia, and the UAE. HDI data was sourced from the UNDP. The error bars represent the upper and lower bounds of the 95% confidence intervals for the proportion of preventable cancers under each scenario.

## Discussion

We found that maximizing the literacy rate to have the highest impact on tobacco prevalence with a projected prevention of 2.2% of all tobacco-related cancer cases. The highest implementation of the MPOWER measures and a 10-unit increase in the tobacco affordability index, each could potentially prevent over 100,000 incident cases in EMR countries by 2050. We also observed that the combined implementation of all anti-tobacco policies is projected to prevent over 400,000 cancers, accounting for 3.1% of all tobacco-related incident cancers in this region. The projected prevention varied by HDI, ranging from 291,425 in low HDI countries to 15,228 in high HDI countries.

In our study, we demonstrated that maximizing literacy rate could be the most effective strategy particularly for countries with medium and low HDI, that may lead to the potential prevention of 1.1%, and 3.9% of all incident tobacco-related cancers, respectively, in these countries by 2050. By improving education, populated countries with low literacy rates (such as Afghanistan, Pakistan, and Yemen) could potentially achieve a notable decrease in smoking prevalence that could lead to the prevention of a considerable number of cancers cases. While increasing literacy is not a targeted anti-tobacco policy, its strong inverse correlation with smoking prevalence has been documented in numerous studies [22–24]. Higher education may facilitate better access to health information and increase the public awareness of the harmful effects of tobacco use [25,26]. Health messages are more likely to be understood and followed by people with higher education [27]. Higher education is also associated with being more aware of the risks of smoking and increases the likelihood of quitting or avoiding smoking [22,28]. Maximizing literacy rates has broader benefits for cancer control beyond just reducing tobacco use undermining the importance of increasing literacy rate in the context of low- and middle-income countries. In such countries, higher education levels additionally lead to a reduction in exposure to other cancer risk factors and improved earlier cancer diagnosis through timely medical referrals [29–32].

We also found that implementing the MPOWER package at its highest level is an effective anti-tobacco policy, potentially preventing around 150,000 tobacco-related cancers in the EMR countries. A similar study conducted in Europe by Gredner and colleagues [4]. demonstrated that the implementation of the MPOWER package could prevent

149,000 lung cancer cases over a 20-year period (2018–2037). Higher prevalence of tobacco smoking in Europe, along with the higher-incidence of lung cancer in both men and women, may have largely accounted for the observed differences. Additionally, the relative risks used for lung cancer in European population were higher than those applied in the present analysis for EMR countries [4]. Moreover, Gredner and colleagues reported a stronger effect of anti-tobacco policies on smoking prevalence in Europe. Their findings indicated that each unit increase in tobacco control policies implementation was associated with a 0.7%–0.9% reduction in prevalence of tobacco smoking across Europe [4]. In contrast, in our study within the EMR, the reduction did not exceed 0.26% per unit increase in the MPOWER score. Although differences in methodological approaches and variations in tobacco policy implementation scoring systems may contribute to the observed differences in policy effects between Europe and the EMR [4], lower public awareness of the harmful effects of tobacco use in EMR countries may pose additional challenges, potentially diminishing the overall effectiveness of anti-tobacco policies and contributing to the lower impact of MPOWER implementation on tobacco smokingprevalence in the region. In the present study, we used a fixed-effects panel regression framework to examine the association between changes in tobacco control policy implementation and smoking prevalence. In contrast, several previous studies particularly from European settings have modeled relative changes in smoking prevalence as a function of changes in policy implementation scores. While informative for short-term behavioral responses, such approaches can be more sensitive to short-term fluctuations and the choice of baseline prevalence. Our level-based modeling strategy provides more stable estimates, is less sensitive to baseline choice, and is better suited to irregular panel data structures, which are common in the EMR context [4,33]. In other regions, particularly the United States, more advanced simulation-based approaches such as smoking history generator models have been used to project the long-term impact of tobacco control policies on smoking prevalence and health outcomes [34]. Although these models can generate highly detailed and potentially more precise projections, they require comprehensive longitudinal smoking profiles, including age- and gender-specific prevalence by smoking status (never, current, and former), age at initiation, cessation rates, and smoking intensity. Such detailed data is not consistently available for most EMR countries.

To further examine the relationship between tobacco control policies, literacy, and smoking prevalence, we conducted a sensitivity analysis assessing the interaction between MPOWER implementation and literacy rate. We observed significant interaction, with the association between MPOWER scores and smoking prevalence attenuated in countries with higher literacy levels. This may reflect higher baseline MPOWER implementation in these settings, leaving less room for improvement, as well as a greater role of compliance, enforcement, and social norms beyond formal policy adoption. Additionally, the effects of MPOWER components are heterogeneous, with taxation showing the strongest impact [35]. However, income growth may offset tax-induced reductions in tobacco affordability, potentially weakening policy effects, particularly in more developed EMR countries.

Most EMR countries have partially implemented various components of the MPOWER package, showing substantial improvement in areas such as health warning labels and monitoring tobacco smoking prevalence. However, several barriers have hindered the full preventive potential of the MPOWER package in this region [9]. These barriers include insufficient compliance with policies like smoke-free public places and bans on indirect tobacco advertisements. Moreover, the development of tobacco cessation centers, a crucial component of the MPOWER package, has seen little progress in most EMR countries [9]. Financial, technical, and human resource limitations have restricted their ability to provide effective pharmacological and social support. Additionally, the lack of proper funding for anti-tobacco initiatives is an important obstacle [9]. Most EMR countries struggle to implement costly policies such as fully funded tobacco cessation programs or to enforce other policies like smoke-free public places and bans on indirect tobacco advertisements, leading to poor compliance and reduced effectiveness [8,9,16]. In such a setting, maximizing MPOWER was identified as the first preventive tobacco prevention priority in the wealthiest countries of the region where the literacy rates are already high and increasing the affordability index may be more challenging due to the high GDP per capita.

Increasing tobacco prices through taxation has been an overlooked anti-tobacco policy in EMR countries. Although tobacco taxation is one of the most cost-effective strategies, most EMR countries have made little progress in this area [36]. As a result, tobacco remains considerably more affordable in the EMR compared to high-income regions like Europe [7]. In 2020, the tobacco affordability index in EMR was estimated at 6.2% of GDP per capita, far lower than most of high-income countries [8]. Over the past decade, most EMR countries have shown some improvement in this area, but only Iran and Yemen have exceeded the acceptable threshold [8]. However, the situation in these two countries was primarily due to economic challenges and a drop in GDP per capita rather than an increase in tobacco taxes [37]. Our results revealed that a 10-unit increase in the tobacco affordability index could perform similarly to the entire MPOWER package, potentially preventing over 100,000 tobacco-related cancers (0.8% of all tobacco-related cancers) in EMRO.

The analysis applied a robust statistical approach and relied on the most reliable available data on policy implementation, tobacco smoking prevalence, and cancer incidence. While MPOWER scores may not fully reflect the implementation of anti-tobacco policies in real-world, they are considered the most comprehensive measures showing the adoption of WHO-FCTC by various countries worldwide aiming to reduce tobacco consumption. Furthermore, while country-specific relative risks by cancer sites were not available for EMR countries, we used site-specific and gender-specific relative risks that were driven from high-quality systematic reviews and meta-analyses that primarily included cohort and case-control studies conducted in various regions of the world and were adjusted for potential confounders. Our analysis quantified statistical uncertainty using a bootstrapping approach. However, the reported confidence intervals do not capture uncertainty related to real-world policy implementation, including feasibility, compliance, political commitment, and enforcement capacity. Consequently, the actual effects of policy implementation may differ from model projections depending on the extent to which real-world implementation aligns with the hypothetical policy scenarios modeled in this study.

We also had several limitations to acknowledge. As with any ecological study, our findings are subject to ecological fallacy. The observed country-level associations should therefore not be interpreted as evidence of individual-level causal relationships between policy measures, smoking behavior, and cancer outcomes. Due to the absence of robust cancer registry data in most EMR countries, we used the GLOBOCAN estimates and thus may have underestimated the incidence rates for some cancers such as lung and larynx in countries with high smoking prevalence such as Iraq, Egypt, Kuwait, and Jordan.

The fixed-effects regression framework used in this study does not consider time-varying unmeasured confounders, including economic growth, enforcement intensity, and broader cultural or social changes. We applied uniform gender-specific policy coefficients across countries, although the impact of tobacco control measures likely varies with factors such as socioeconomic status, baseline smoking prevalence, and MPOWER implementation levels and considering both year and country absorb some part within country variation. More advanced approaches such as age–period–cohort models or smoking simulation models (e.g., smoking history generators), which require more detailed information on smoking behaviors may provide deeper insights into future smoking prevalence and tobacco-attributable cancer burden in the EMR.

We assumed that the effect of improvements in policy implementation was immediate, beginning in 2025 and remaining constant over the subsequent 25 years. In practice, however, tobacco control policies are typically implemented gradually, and behavioral responses often occur with time lags. However, due to the calendar years of the available data on tobacco smoking prevalence and policy interventions in EMR countries, we were unable to apply formal lagged analyses.

We acknowledge that a proportion of the observed prevalence may reflect the use of other tobacco products, such as water-pipe smoking, which may be associated with different relative risks for certain cancer sites compared with cigarette smoking. However, findings from our previous study in the EMR showed that only about 1% of cancer cases were attributable to water-pipe smoking [6]. Therefore, using tobacco smoking may not impose substantial effect on our

PLOS Medicine

estimates, while it has considered all types of tobacco smoked in the region. In addition, MPOWER scores, literacy levels, and affordability indices may be affected by measurement error and may not fully reflect the real-world implementation or enforcement of policies. Such classical measurement error in the explanatory variables is likely to bias effect estimates toward the null, resulting in potentially conservative assessments of policy impact on smoking prevalence and cancer prevention.

Our study indicates that the implementation of anti-tobacco policies can provide important reductions in tobacco smoking prevalence in the EMR region, with the potential of preventing 442,292 tobacco-related cancers during the next 25 years. We found that improving literacy rates was more important than implementing targeted anti-tobacco policies in countries with low and medium human development index. In contrast, targeted anti-tobacco policies remained important in countries with high human development index. These results highlight the importance of considering country-specific conditions to optimize prevention of smoking and tobacco-related cancer in the EMR region.

## Supporting information

**S1 Table. Gender-specific Tobacco smoking prevalence, MPOWER index score, cigarette affordability index, and literacy rate for each EMR country.**
(DOCX)

**S2 Table. The association between change in MPOWER, affordability index, literacy rate, and prevalence of tobacco smoking in the EMR countries using historical data from 2010 to 2020.**
(DOCX)

**S3 Table. Gender -specific Current prevalence of tobacco smoking, prevalence of tobacco smoking with the highest MPOWER, prevalence with a 10-unit increase in affordability index, prevalence with maximizing literacy rate, and prevalence of tobacco smoking under all policies combined scenario in each EMR country.**
(DOCX)

**S4 Table: Projected number and proportion of all incident tobacco-related cancers, for the years 2025–2050, that would be attributable to current tobacco smoking in EMR countries.**
(DOCX)

**S5 Table. Relative risks for the associations between current tobacco smoking and associated cancer types.**
(DOCX)

**S6 Table. Projected number and proportion of all tobacco-related cancers and cancers attributable to current tobacco smoking in EMR countries stratified by cancer site (2025–2050).**
(DOCX)

**S7 Table. Estimated number and proportion of preventable cancer cases attributable to each policy measure for each country stratified by gender and level of Human Development Index (HDI) in the adult population of EMR in 2025–2050.**
(DOCX)

**S8 Table. Number of projected preventable cancers by 2050, that could be achieved though highest MPOWER implementation, a 10-unit increase in tobacco affordability index, maximizing literacy rate, and combined implementations of all policies in EMR countries assuming cancer incidence is 10% higher than GLOBOCAN estimates.**
(DOCX)

**S9 Table. Number of avoidable cancers due to highest MPOWER, a 10-unit increase in affordability index, maximizing level of literacy, and all combined policies in EMR countries (2025–2050) stratified by cancer site assuming cancer incidence is 10% higher than GLOBOCAN estimates.**
(DOCX)

**S10 Table. Number of projected preventable cancers by 2050, that could be achieved though highest MPOWER implementation, a 10-unit increase in tobacco affordability index, maximizing literacy rate, and combined implementations of all policies in EMR countries assuming cancer incidence is 10% lower than GLOBOCAN estimates.**
(DOCX)

**S11 Table. Number of avoidable cancers due to highest MPOWER, a 10-unit increase in affordability index, maximizing level of literacy, and all combined policies in EMR countries (2025–2050) stratified by cancer site assuming cancer incidence is 10% lower than GLOBOCAN estimates.**
(DOCX)

**S12 Table. Fixed-effects multiple linear regression examining the association between changes in tobacco smoking prevalence and policy interventions, including MPOWER score, literacy rate, and affordability index, with a multiplicative interaction between literacy rate and MPOWER score.**
(DOCX)

**S13 Table. Projected number and proportion of preventable tobacco-related cancers under the combined policy scenario accounting for interaction between MPOWER score and literacy rate, A 25-year projection from 2025–2050 stratified by country and gender.**
(DOCX)

**S14 Table. Projected number and proportion of preventable tobacco-related cancers under the combined policy scenario *accounting* for interaction between MPOWER score and literacy rate, A 25-year projection from 2025–2050 stratified by cancer site and gender.**
(DOCX)

## Acknowledgments

We would like to thank our colleagues, Elnaz Saeedi and Pedram Fattahi, at the Cancer Institute of Iran for their valuable support.

**Disclosure:** Where authors are identified as personnel of the International Agency for Research on Cancer/World Health Organization, the authors alone are responsible for the views expressed in this article and they do not necessarily represent the decisions, policy or views of the International Agency for Research on Cancer/World Health Organization.

## Author contributions

**Conceptualization:** Saeed Nemati, Mojtaba Vand Rajabpour, Mattias Johansson.

**Data curation:** Xiaoshuang Feng, Farrokh Heidari.

**Formal analysis:** Saeed Nemati, Xiaoshuang Feng, Mahdi Sheikh.

**Investigation:** Mojtaba Vand Rajabpour, Negar Taheri, Farrokh Heidari, Ebrahim Karimi, Sepideh Abdi.

**Methodology:** Saeed Nemati, Mojtaba Vand Rajabpour, Xiaoshuang Feng, Negar Taheri, Harriet Rumgay, Ebrahim Karimi, Mattias Johansson, Mahdi Sheikh.

**Project administration:** Mojtaba Vand Rajabpour, Mahdi Sheikh.

**Resources:** Mahdi Sheikh.

**Software:** Ebrahim Karimi.

**Supervision:** Mojtaba Vand Rajabpour, Harriet Rumgay, Mattias Johansson, Mahdi Sheikh.

**Validation:** Sepideh Abdi, Mattias Johansson, Mahdi Sheikh.

**Visualization:** Saeed Nemati, Negar Taheri, Sepideh Abdi.

**Writing – original draft:** Saeed Nemati, Farrokh Heidari, Ebrahim Karimi.

**Writing – review & editing:** Mojtaba Vand Rajabpour, Xiaoshuang Feng, Negar Taheri, Harriet Rumgay, Sepideh Abdi, Mattias Johansson, Mahdi Sheikh.

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
