## [Editor Report · Decision Letter 0]

18 Sep 2025

Dear Dr Nemati,

Thank you for submitting your manuscript entitled "Projected Impact of Tobacco Control Policies on Cancer Prevention in the Eastern Mediterranean Region, 2025 to 2050" for consideration by PLOS Medicine.

Your manuscript has now been evaluated by the PLOS Medicine editorial staff and I am writing to let you know that we would like to send your submission out for external peer review.

Before we can send your manuscript out for review, please revise the Methods section to include comprehensive details on the model. Currently, the Methods section lacks key details, and presenting the modeling methods in the Appendix is insufficient. Please revise it prior to resubmission.

Please re-submit your manuscript within two working days, i.e. by Sep 22 2025.

Feel free to email me at atosun@plos.org or us at plosmedicine@plos.org if you have any queries relating to your submission.

Kind regards,

Alexandra Tosun, PhD

Senior Editor

PLOS Medicine

---

## [Decision Letter · Decision Letter 1]

2 Dec 2025

Dear Dr Nemati,

Many thanks for submitting your manuscript "Projected Impact of Tobacco Control Policies on Cancer Prevention in the Eastern Mediterranean Region, 2025 to 2050" (PMEDICINE-D-25-03248R1) to PLOS Medicine. The paper has been reviewed by a subject expert, academic editor and a statistician; their comments are included below and can also be accessed here: [LINK]

As you will see, the reviewers find the work of interest but have raised extensive comments pertaining to the analyses and some of the conclusions. After discussing the paper with the editorial team and an academic editor with relevant expertise, I'm pleased to invite you to revise the paper in response to the reviewers' comments. We plan to send the revised paper to some or all of the original reviewers, and we cannot provide any guarantees at this stage regarding publication. Please be advised that we may seek the input of an additional independent reviewer on a revised manuscript.

We ask that you submit your revision by Dec 23 2025 11:59PM. However, if this deadline is not feasible, please contact me by email, and we can discuss a suitable alternative.

Don't hesitate to contact me directly with any questions (atosun@plos.org).

Best regards,

Alexandra

Alexandra Tosun, PhD

Senior Editor

PLOS Medicine

atosun@plos.org

Comments from the reviewers:

Reviewer #1: The paper addresses an important question regarding the projected impact of tobacco control policies on cancer incidence in the Eastern Mediterranean Region (EMR). While the study is timely and well-motivated, several methodological and statistical aspects need clarification or additional justification to ensure transparency and credibility of the projections.

Major Comments

a) The analysis implicitly assumes a causal relationship between tobacco control measures (MPOWER score, affordability index, and literacy rate) and smoking prevalence. However, the fixed-effects linear regression framework can only identify associations, not causality. This limitation should be explicitly acknowledged, with a discussion of potential time-varying confounders such as economic growth, enforcement intensity, or cultural changes.

b) The regression models are based on six time points (2010-2020), which may not adequately capture non-linear trends or policy lag effects. The linearity assumption should be justified, and alternative specifications (e.g., random-effects, mixed, or spline-based models) explored. Moreover, the methods do not explain how uncertainty was propagated from regression coefficients through to PAF and PIF estimates. It should be clarified whether bootstrapping, Monte Carlo simulation, or the delta method was used to derive confidence intervals.

c) Policy interventions rarely have immediate effects. The authors should clarify whether lag terms were considered (e.g., policy effects at year t influencing smoking prevalence at t+2). Without lag adjustment, short-term associations may overestimate the projected preventive potential.

d) Including both country and year fixed effects can control for unobserved heterogeneity but may also absorb much of the true policy variation, particularly when changes in MPOWER or literacy are slow. This limitation should be discussed, and measures of model fit (R² within, between, and overall) reported for transparency.

e) Five countries were excluded due to missing data. The authors should examine whether these exclusions bias results (e.g., systematically low-HDI or high-prevalence countries). Sensitivity analyses using imputed or regional averages for missing indicators would strengthen confidence in regional estimates.

f) The method of multiplying "policy gaps" by regression coefficients assumes constant marginal effects and independence among policies. This simplification may not hold if policies interact—for instance, literacy and affordability could jointly modify smoking behavior. A sensitivity analysis exploring non-additivity of combined policies is recommended.

g) GLOBOCAN projections are useful but may not reflect national registry differences or population dynamics. The authors should justify using GLOBOCAN rather than national data or age-period-cohort projections and discuss potential biases this introduces.

h) The use of Levin's formula assumes uniform exposure and constant relative risk across populations. Given heterogeneity in smoking intensity and patterns across EMR countries, this assumption should be acknowledged. The manuscript should also clarify whether relative risks were applied uniformly across all countries or varied by gender and cancer site (as suggested in Supplementary Table 5).

i) The method for deriving confidence intervals for projected cancer cases remains unclear. It should be explained whether uncertainty in smoking prevalence, relative risks, and incidence projections was jointly propagated; otherwise, reported intervals may be overly narrow.

j) HDI stratification is a useful approach but was implemented categorically, which may obscure within-group heterogeneity. The authors could justify the grouping scheme or test HDI as a continuous variable in sensitivity analysis.

Minor Comments

a) Ensure consistent notation throughout (e.g., β₁ vs. B₁, consistent equation formatting).

b) Provide a clear rationale for the 10-unit increase in the affordability index—does it reflect a realistic or policy-relevant shift?

c) Clarify whether sex-specific models were estimated separately or through interaction terms, and whether the results were statistically compared.

d) Indicate if all variables were standardized before regression to aid interpretability.

e) Specify how 2025 prevalence was derived and whether demographic projections were incorporated into subsequent estimates.

f) A directed acyclic graph (DAG) or conceptual framework linking policy → smoking → cancer would strengthen the methodological transparency.

g) Sensitivity analyses using alternative relative risks or cancer projections (e.g., varying GLOBOCAN assumptions by ±10%) would enhance robustness.

h) Quantify uncertainty in cancer incidence projections due to possible under-reporting in registry data.

i) Consistently describe scaling of variables: for example, clarify whether a higher affordability index implies higher or lower affordability.

j) Explain the MPOWER scoring system and interpret the meaning of a one-unit change in the regression context.

k) Check for multicollinearity among policy variables (e.g., literacy and MPOWER) using VIF or correlation analysis, and report results.

l) Clarify whether regressions were population-weighted; unweighted models may overrepresent small countries.

m) Note that projecting linear effects to 2050 assumes policy impacts remain stable over 30 years—this should be explicitly acknowledged as a limitation.

n) Ensure all acronyms (MPOWER, PAF, PIF) are defined at first mention.

o) Discuss data harmonisation across sources (WHO, World Bank, IARC) and potential measurement inconsistencies.

p) Justify using the World Bank's HDI instead of the standard UNDP classification.

q) Ensure the equations and tables display appropriate spacing and consistent notation.

r) Review confidence intervals in Tables 1-2, which appear narrower than expected given multi-step uncertainty; verify computation.

s) Temper the causal interpretation that "improving literacy offers greater benefit than MPOWER," as differing mechanisms and potential biases limit comparability.

Reviewer #2: Overall I find this an interesting paper and highly relevant to current global policy priorities. My comments are below:

1) Make it clear that "increase in affordability index" means lower affordability and give a brief sentence explaining how it is measured for readers' convenience to aid interpretation of your findings.

2) Your analysis appears to assume immediate effect of policy post 2025. First, confirm whether this is the case, and second, perhaps discuss how this is an optimistic scenario and how tobacco control policies are often phased in adoption and implementation.

3) Perhaps you may consider analysing multiple scenarios, including more realistic scenarios. For example you could take one country from 2000 to 2025 with a more "successful" implementation of MPOWER and apply that (i.e. what could other countries realistically achieve if they emulated this specific country?) to the others.

4) Explain how you calculated confidence intervals for your PIFs. You may consider use of Markov chain analysis to obtain confidence intervals

5) You may consider discussing latency effects (although you touch on this at the end of the manuscript by hinting at the possibility of conducting age-period-cohort analyses) in the association between smoking and cancer outcomes. For example preventing uptake among younger people will only result in reductions in mortality decades later as the cancer outcomes you analysed take many years to develop when caused by smoking.

6) I suggest reviewing and discussing other literature relevant to other WHO regions, in particular whether other researchers have attempted projecting impact of tobacco control policies, how your methods and findings may have differed, and whether there are any strengths or limitations of your approach compared with other studies.

7) Literacy may not be a confounder for the association between tobacco control policies and smoking prevalence, or simply a direct determinant. Literacy may also interact with MPOWER policies, acting as a moderator of policy effectiveness (for example literacy may represent a proxy for health literacy, which in turn influences the effectiveness of tobacco control information campaigns at changing behaviour). Perhaps you may briefly discuss how and whether the association between literacy and smoking prevalence is causal, and potential pathways.

8) You may speculate on whether individual MPOWER domains may be driving changes in smoking prevalence more than others.

9) You may discuss the strengths and limitations of studies/sources of RR estimates used in your analysis (e.g. biases, study design limitations, generalisability to EMR etc).

10) Likewise, are there any limitations as to how MPOWER scores are estimated?

11) You model also seems to assume the association between affordability and smoking prevalence is linear; you may consider fitting a quadratic term in your model to asses this or briefly comment on this in the discussion.

12) In your introduction you mention alternative forms of tobacco products such as water pipes. Are there included in the smoking prevalence estimates? Could excluding these products or not disaggregating them from cigarettes be a limitation of your study? For example do these have different RRs for their associations with different cancers?

13) Please specify the age ranges for your tobacco use prevalence estimates by country.

---

* Please upload any figures associated with your paper as individual TIF or EPS files with 300dpi resolution at resubmission; please read our figure guidelines for more information on our requirements: http://journals.plos.org/plosmedicine/s/figures. While revising your submission, we strongly recommend that you use PLOS's NAAS tool (https://ngplosjournals.pagemajik.ai/artanalysis) to test your figure files. NAAS can convert your figure files to the TIFF file type and meet basic requirements (such as print size, resolution), or provide you with a report on issues that do not meet our requirements and that NAAS cannot fix.

After uploading your figures to PLOS's NAAS tool - https://ngplosjournals.pagemajik.ai/artanalysis, NAAS will process the files provided and display the results in the "Uploaded Files" section of the page as the processing is complete.

If the uploaded figures meet our requirements (or NAAS is able to fix the files to meet our requirements), the figure will be marked as "fixed" above. If NAAS is unable to fix the files, a red "failed" label will appear above.

When NAAS has confirmed that the figure files meet our requirements, please download the file via the download option, and include these NAAS processed figure files when submitting your revised manuscript.

* Please ensure that your that your Data Availability Statement, Competing Interests statement and Funding statement are complete. Please include the funders' URLs in the statement.

* Please ensure that the study is reported according to the appropriate guideline and include the completed checklist as Supporting Information. When completing the checklist, please use section and paragraph numbers, rather than page numbers. Please add the following statement, or similar, to the Methods: "This study is reported as per [XXXX] guideline (S1 Checklist)."

FIGURES AND TABLES

SUPPLEMENTARY MATERIAL

REFERENCES

OBSERVATIONAL STUDIES

* Abstract: Please include the study design, population and setting, number of participants, years during which the study took place (enrollment and follow up), length of follow up, and main outcome measures.

* Please ensure that the study is reported according to the STROBE (or appropriate STOBE extension) guideline (available from: https://www.equator-network.org/reporting-guidelines/strobe) and include the completed STROBE (or STROBE extension) checklist as Supporting Information. Please add the following statement, or similar, to the Methods: "This study is reported as per the Strengthening the Reporting of Observational Studies in Epidemiology (STROBE) guideline (S1 Checklist)." When completing the checklist, please use section and paragraph numbers, rather than page numbers.

* [FOR POPULATION HEALTH/REGISTRY STUDIES] Please ensure that the study is reported according to the RECORD guideline (available from https://www.record-statement.org) and include the completed checklist as Supporting Information. Please add the following statement, or similar, to the Methods: "This study is reported as per the Reporting of Studies Conducted using Observational Routinely-Collected Data (RECORD) guideline (S1 Checklist)." When completing the checklist, please use section and paragraph numbers, rather than page numbers.

* [FOR POPULATION HEALTH ESTIMATES] Please ensure that the study is reported according to the GATHER statement (available from https://www.equator-network.org/reporting-guidelines/gather-statement) and include the completed checklist as Supporting Information. Please add the following statement, or similar, to the Methods: "This study is reported as per the Guidelines for Accurate and Transparent Health Estimates Reporting (GATHER) statement (S1 Checklist)." When completing the checklist, please use section and paragraph numbers, rather than page numbers.

* [FOR MEDIATION ANALYSES] We recommend that the study is reported according to the AGReMA statement (https://agrema-statement.org/#:~:text=AGReMA%20is%20an%20evidence%2D%20and,randomised%20trials%20and%20observational%20studies) and include the completed checklist as Supporting Information. Please add the following statement, or similar, to the Methods: "This study is reported as per the Guideline for Reporting Mediation Analyses (AGReMA) statement (S1 Checklist)." When completing the checklist, please use section and paragraph numbers, rather than page numbers.

* For all observational studies, in the manuscript text, please indicate: (1) the specific hypotheses you intended to test, (2) the analytical methods by which you planned to test them, (3) the analyses you actually performed, and (4) when reported analyses differ from those that were planned, transparent explanations for differences that affect the reliability of the study's results. If a reported analysis was performed based on an interesting but unanticipated pattern in the data, please be clear that the analysis was data driven.

* Please state in the Methods section whether the study had a prospective protocol or analysis plan. If a prospective analysis plan (from your funding proposal, IRB or other ethics committee submission, study protocol, or other planning document written before analyzing the data) was used in designing the study, please include the relevant document(s) with your revised manuscript as a Supporting Information file to be published alongside your study and cite it in the Methods section. A legend for this file should be included at the end of your manuscript. If no such document exists, please make sure that the Methods section transparently describes when analyses were planned, and when/why any data-driven changes to analyses took place. Changes in the analysis, including those made in response to peer review comments, should be identified as such in the Methods section of the paper, with rationale.

MODELLING STUDIES

The following list is derived from Geoffrey P Garnett, Simon Cousens, Timothy B Hallett, Richard Steketee, Neff Walker. Mathematical models in the evaluation of health programmes. (2011) Lancet DOI:10.1016/S0140-6736(10)61505-X:

* If pertinent, please provide a diagram that shows the model structure, including how the natural history of the disease is represented, the process and determinants of disease acquisition, and how the putative intervention could affect the system.

* Please provide a complete list of model parameters, including clear and precise descriptions of the meaning of each parameter, together with the values or ranges for each, with justification or the primary source cited and important caveats about the use of these values noted.

* Please provide a clear statement about how the model was fitted to the data, including goodness-of-fit measure, the numerical algorithm used, which parameter varied, constraints imposed on parameter values, and starting conditions.

* For uncertainty analyses, please state the sources of uncertainties quantified and not quantified [can include parameter, data, and model structure].

* Please provide sensitivity analyses to identify which parameter values are most important in the model. Uncertainty estimates seek to derive a range of credible results on the basis of an exploration of the range of reasonable parameter values. The choice of method should be presented and justified.

* Please discuss the scientific rationale for the choice of model structure and identify points where this choice could influence conclusions drawn. Please also describe the strength of the scientific basis underlying the key model assumptions.

* For studies that develop a prediction model or evaluate its performance, please ensure that the study is reported according to the TRIPOD statement (https://www.equator-network.org/reporting-guidelines/tripod-statement) and include the completed checklist as Supporting Information. Please add the following statement, or similar, to the Methods: "This study is reported as per the Transparent Reporting of a Multivariable Prediction Model for Individual Prognosis Or Diagnosis (TRIPOD) statement (S1 Checklist)." For studies using machine learning, please use the TRIPOD-AI checklist. When completing the checklist, please use section and paragraph numbers, rather than page numbers.

---

## [Decision Letter · Decision Letter 2]

13 Feb 2026

Dear Dr. Nemati,

Thank you very much for re-submitting your manuscript "Projected Impact of Tobacco Control Policies on Cancer Prevention in the Eastern Mediterranean Region, 2025 to 2050" (PMEDICINE-D-25-03248R2) for review by PLOS Medicine.

Thank you for your detailed response to the reviewers' comments. I have discussed the paper with my colleagues, with the academic editor with relevant expertise, and it has also been seen again by the original reviewers. The reviewers were mostly satisfied with the changes made to the paper, but they provided additional comments that require a careful response and potential changes/additions to your analysis. Please address the reviewers' and editors' comments in a further revision. When submitting your revised paper, please once again include a detailed point-by-point response to the editorial comments. The remaining issues that need to be addressed are listed at the end of this email.

We ask that you submit your revision by Feb 20 2026. However, if this deadline is not feasible, please contact me (atosun@plos.org) or the journal staff by email, and we can discuss a suitable alternative.

We look forward to receiving the revised manuscript.

Sincerely,

Alexandra Tosun, PhD

Senior Editor

PLOS Medicine

plosmedicine.org

Comments from Reviewers:

Reviewer #1: Thank you for thoroughly addressing all my comments and for providing detailed clarifications in the revised manuscript. I appreciate the considerable effort you have invested in enhancing the rigor and transparency of the analyses. The following are additional minor comments that may further improve clarity and interpretation of the results:

a) While regression coefficients are interpreted as percentage changes in smoking prevalence, it would be helpful to briefly contextualise these effects in absolute prevalence terms (e.g., percentage-point changes). This may aid policy interpretation, particularly when translating reductions in smoking prevalence into projected cancer cases.

b) MPOWER scores, literacy rates, and affordability indices are subject to measurement error and imperfect policy enforcement. The authors may wish to acknowledge that classical measurement error in these explanatory variables could attenuate estimated associations, potentially resulting in conservative estimates of policy impact.

c) Although already implicit, the manuscript would benefit from a short, explicit statement noting that ecological associations at the country level may not directly reflect individual-level causal mechanisms (i.e., the ecological fallacy), particularly when linking policy measures to smoking behaviour and cancer outcomes.

d) It would be useful to clarify whether projected reductions in smoking prevalence were constrained to avoid implausible negative prevalence values in countries with low baseline smoking rates under combined policy scenarios.

e) The manuscript could benefit from a brief conceptual distinction between statistical uncertainty (captured via the bootstrapping procedure) and policy implementation uncertainty (e.g., feasibility, compliance, and enforcement), which is not formally modelled but is important for interpretation of the findings.

Reviewer #2: I believe that the authors have adequately addressed the reviewers' comments within the scope of this paper, and that the changes made to the manuscript have significantly improved it.

I would only make two final optional comments:

1) You said "Our analysis identified a statistically significant interaction between literacy rate and MPOWER score, indicating that these factors jointly modify smoking behavior. Specifically, the effect of the MPOWER score was attenuated at higher literacy levels...". Are you able to explain why your model may have shown this finding? Is this counterintuitive? Can this result be contextualised by findings of previous research?

2) Regarding my original comment #3 (i.e. modelling "more realistic scenarios") I believe your approach makes sense. However, another approach my had in mind was to select a country that could be considered a "leader" in the Region (say Qatar as an example) and that in this scenario all countries' MPOWER scores are raised to that of the "leader" to address the question of what if all countries of the Region emulated the example of best practice set by this leading country, and then later discuss which MPOWER domains most countries should address to catch up with the "leader".

Requests from Editors:

Please note that your point-by-point response did not address the list of editorial requests included in the prior decision letter. Please consider these points when revising your manuscript. In your response to this decision letter, please address the points under "OBSERVATIONAL STUDIES" and "MODELLING STUDIES" from the previous letter.

GENERAL

* Please confirm that your title complies with to PLOS Medicine's style. Your title must be nondeclarative and not a question. It should begin with main concept if possible. "Effect of" (or “Impact of”) should be used only if causality can be inferred, i.e., for an RCT. Please place the study design ("A randomized controlled trial," "A retrospective study," "A modelling study," etc.) in the subtitle (ie, after a colon).

* Statistical reporting: Please revise throughout the manuscript, including tables and figures.

- Please report statistical information as follows to improve clarity for the reader, ""XX% (95% CI [XX,YY]; p</=)"".

- Please separate upper and lower bounds with commas instead of hyphens as the latter can be confused with reporting of negative values.

- Please repeat statistical definitions (HR, CI etc.) for each set of parentheses.

* Please ensure that all abbreviations are defined at first use throughout the text (including statistical abbreviations).

* Please ensure that tables and figures, including those in supplementary files, are appropriately referenced in the main text.

* Please review your text for claims of novelty or primacy (e.g. 'for the first time' or ‘novel’) and remove this language.

* Please confirm that any use of statistical terms (such as trend or significant) are supported by the data, and if not please remove them. The term trend should be used only when the test for trend has been conducted.

* Please define all acronyms used in each figure or table in the corresponding legend.

* Please include all relevant links and access details in the Data Availability Statement. A reference to the Appendix is not sufficient.

* The terms gender and sex are not interchangeable (as discussed in https://www.who.int/health-topics/gender#tab=tab_1 ); please use the appropriate term.

ABSTRACT

* Please confirm that your abstract complies with our requirements, including providing all the information relevant to this study type https://journals.plos.org/plosmedicine/s/submission-guidelines#loc-abstract

* Please confirm that all numbers presented in the abstract are present and identical to numbers presented in the main manuscript text.

* In the abstract, please include the important dependent variables that are adjusted for in the analyses (if applicable).

* Please ensure that all abbreviations used are defined at first use.

* Please provide a brief explanation what the cigarette affordability index is or means (e.g.: higher values of the affordability index equal lower cigarette affordability).

* Please mention that “maximized literacy rates” means 100%.

* Please provide CI values for all numerical results presented.

AUTHOR SUMMARY

* Please include a short, non-technical Author Summary of your research to make findings accessible to a wide audience that includes both scientists and non-scientists. The Author Summary should immediately follow the Abstract in your revised manuscript. This text is subject to editorial change and should be distinct from the scientific abstract. Ideally each sub-heading should contain 2-3 single sentence, concise bullet points containing the most salient points from your study. In the final bullet point of 'What Do These Findings Mean?', please include the main limitations of the study in non-technical language. Please see our author guidelines for more information: https://journals.plos.org/plosmedicine/s/revising-your-manuscript#loc-author-summary.

INTRODUCTION

* “in more than 13 anatomical sites”. – We think it’d be useful to list the specific cancer types here.

* “While the maximum score for the ‘M’ component is 4, the other six components each have a maximum score of 5, resulting in an overall MPOWER score ranging from 7 to 34 (13).” – Please include this information in the Methods section.

METHODS AND RESULTS

* Please remove all numbering from headers.

* “aged older than 15” – when reporting age, please add a unit, such as ‘years’.

* “We also evaluated the presence of multiplicative interactions between policy measures. Our analysis identified…” – Please report findings of the sensitivity analyses in the Result section, not the Methods.

* “Increasing the tobacco affordability index, as a second alternative scenario, was projected to prevent 0.8% (95% CI: 0.6 – 0.9) of all incident tobacco-related cancers in the EMR.” – please provide absolute numbers as well.

* Figure 2 and 3: The figures require revision:

1) Please ensure that the error bars do not overlap with the numbers on the graph.

2) Please provide a description and unit for the x-axis.

3) In the figure description, please define the meaning of the error bar.

* Figure 4: The figure requires revision:

1) Please provide a unit for the y-axis.

2) Please note that there’s a typo in “Medium HDI”.

3) Please include the error bars and please define their meaning in the figure description.

4) In the figure description, please provide definitions of low, medium and high HDI.

5) Does during the next 25 years mean in 25 years (i.e. in 2050)?

* Please confirm that you specified the variables controlled for in all relevant Tables.

* Please confirm that you provided the unadjusted comparisons as well as the adjusted comparisons in all relevant Tables (if applicable).

DISCUSSION

* “Additionally, the relative risks used for lung cancer in the EU were significantly higher than those estimated in Europe (33).” – please clarify.

* Please refer to high income countries rather than "developed" or "Western" countries.

* Please remove the 'conclusions' subheading from the discussion. Please also remove any other subheadings from the discussion.

General Editorial Requests

---

## [Decision Letter · Decision Letter 3]

18 Mar 2026

Dear Dr Nemati,

On behalf of my colleagues and the Academic Editor, Emily Banks, I am pleased to inform you that we have agreed to publish your manuscript "Tobacco Control Policies on Cancer Prevention in the Eastern Mediterranean Region, 2025 to 2050: A Modeling Study" (PMEDICINE-D-25-03248R3) in PLOS Medicine.

I appreciate your thorough responses to the reviewers' and editors' comments throughout the editorial process. For transparency, I am sharing the statistical reviewer's feedback on your final responses. We look forward to publishing your manuscript, and editorially there are only a few remaining points that should be addressed prior to publication. We will carefully check whether the changes have been made. If you have any questions or concerns regarding these final requests, please feel free to contact me at atosun@plos.org.

Please see below the minor points that we request you respond to:

1) Abstract: Please note that there is one instance where you used a hyphen instead of a comma to separate the upper and lower boundaries. Also, please define HDI the first time it is used.

2) Figure 3: In (C), two grey lines appear (larynx and lung) that don’t seem to belong to the graph. Additionally, the error bar for "Lung" in (D) appears to have a different format and is missing a cap on the right side.

3) We suggest including your response to Reviewer #2's comment #1 (regarding the literacy rate) in the discussion. Feel free to shorten it.

PRESS

Sincerely,

Alexandra Tosun, PhD

Senior Editor

PLOS Medicine

Comments from Reviewers:

Reviewer #1: Thank you for thoroughly addressing all my comments and providing detailed clarifications in your revised manuscript. I appreciate the effort you have put into enhancing the rigor and transparency of your analyses.

The manuscript now reflects a robust and valuable contribution to subject, I am happy to recommend this work for publication and look forward to seeing its positive impact in the field.